# Serological Data Shows Low Levels of Chikungunya Exposure in Senegalese Nomadic Pastoralists

**DOI:** 10.3390/pathogens8030113

**Published:** 2019-07-27

**Authors:** Mame Cheikh Seck, Aida Sadikh Badiane, Julie Thwing, Delynn Moss, Fatou Ba Fall, Jules Francois Gomis, Awa Bineta Deme, Khadim Diongue, Mohamed Sy, Aminata Mbaye, Tolla Ndiaye, Aminata Gaye, Yaye Die Ndiaye, Mamadou Alpha Diallo, Daouda Ndiaye, Eric Rogier

**Affiliations:** 1Department of Parasitology, Faculty of Medicine and Pharmacy, Cheikh Anta Diop University, Dakar 12500, Senegal; 2Malaria Branch, Division of Parasitic Diseases and Malaria, Centers for Disease Control and Prevention, Atlanta, GA 30333, USA; 3President’s Malaria Initiative, Atlanta, GA 30303, USA; 4Division of Foodborne, Waterborne, and Environmental Diseases, National Center for Emerging and Zoonotic Infectious Diseases, Centers for Disease Control and Prevention, Atlanta, GA 30333, USA; 5Senegal National Malaria Control Program, Dakar 999066, Senegal

**Keywords:** serology, chikungunya, seroprevalence, nomadic pastoralists, Senegal, multiplex bead assay

## Abstract

The chikungunya virus (CHIKV) is spread by *Aedes aegypti* and *Ae. albopictus* mosquitos worldwide; infection can lead to disease including joint pain, fever, and rash, with some convalescent persons experiencing chronic symptoms. Historically, CHIKV transmission has occurred in Africa and Asia, but recent outbreaks have taken place in Europe, Indonesia, and the Americas. From September to October 2014, a survey was undertaken with nomadic pastoralists residing in the northeast departments of Senegal. Blood dried on filter paper (dried blood spots; DBS) were collected from 1465 participants of all ages, and assayed for Immunoglobulin G (IgG) antibodies against CHIKV E1 antigen by a bead-based multiplex assay. The overall seroprevalence of all participants to CHIKV E1 was 2.7%, with no persons under 10 years of age found to be antibody positive. Above 10 years of age, clear increases of seroprevalence and IgG levels were observed with increasing age; 7.6% of participants older than 50 years were found to be positive for anti-CHIKV IgG. Reported net ownership, net usage, and gender were all non-significant explanatory variables of seropositivity. These data show a low-level historical exposure of this pastoralist population to CHIKV, with no evidence of recent CHIKV transmission in the past decade.

## 1. Introduction

The chikungunya virus (CHIKV) belongs to the genus *Alphavirus* (family *Togaviridae*), and is transmitted to humans primarily by *Aedes aegypti* and *Ae. albopictus* mosquitos, although it has also been found in other mosquitos [1,2,3,4]. CHIKV infection can present as asymptomatic, but the majority of infected persons will develop symptoms such as headache, fever, myalgia, and moderate to severe joint pains [5,6]. CHIKV was first isolated from a Tanzanian patient with dengue-like symptoms in the early 1950s [7], and has been confirmed on the continent in numerous African studies since the initial report [8,9,10]. Besides attempting to identify persons with active infection, seroepidemiological studies have also shed light on individual- and population-level exposure to CHIKV, with the added advantage of expanding the window of time of finding a “positive” through antibody detection [11,12]. In humans, IgG responses are known to occur to CHIKV E1 and E2 antigens, which are attractive for serological studies [12,13,14,15].

The CHIKV was identified in Senegal in the 1960s [16,17], with defined outbreaks occurring in 2009, 2010, and 2015, and a known sylvatic cycle with infected monkeys identified in multiple studies [18,19,20,21]. Recent studies have confirmed current CHIKV presence in Senegal both by detection of active viremia and serological evidence [19,20,22]. However, almost all historical and even recent CHIKV studies have been conducted in the relatively tropical and sparsely populated southeastern zone of Senegal [19,23,24]. As of early 2019, no published studies could be identified that had investigated population exposure to CHIKV of persons living in the dry northern part of Senegal. Furthermore, epidemiological studies of infectious disease exposure in Senegalese nomadic populations is largely nonexistent.

For the current study, populations of nomadic pastoralists were sampled in Senegal in 2014 for an integrated seroepidemiological study. This population has more permanent dwellings in the north and central regions and spends the dry season in the south in order to seek appropriate grazing lands for their cattle, which is dependent on the rainy and dry seasons [25]. This report outlines the findings regarding anti-CHIKV IgG antibodies found in this study population.

## 2. Results

### 2.1. Study Population and Sampling Locations in Senegal

From September to October 2014, participant enrollment occurred in five districts in the northeastern regions of Senegal: Dagana, Podor, Pété, Ranérou, and Kanel (Figure 1). Participant age ranged from 1 to 80 years, and 43.5% were female. In total, 1465 persons provided a dried blood spot (DBS) sample to allow for serological data collection of anti-CHIKV IgG antibodies. Of these, 1463 (99.9%) provided valid serology data as described in Methods.

### 2.2. Range of IgG Responses to CHIKV E1 Antigen and Seropositivity Definition

From the fluorescence signal of the bead-based IgG detection assay (median fluorescence intensity minus background, MFI-bg), a range of signal intensities was observed for the blood samples from this population (Figure 2). Log-transformation of these MFI-bg data showed a clear unimodal population of low signal intensities (under 200 MFI-bg), with few individuals showing an assay signal above this level. A two-component finite mixture model (FMM) was used to allow maximum likelihood estimate (MLE) predictions to determine if two unique populations existed, and outputs illustrated two components with a statistically significant difference in the means. The two-component FMM had a very good fit to the data, with a z value of 62.6 for fitting both to the first and second component; both *p*-values <0.0001. The lognormal MFI-bg mean for the first, leftmost population was 15.6 (95% confidence interval (CI): 0.0–308.0), and that for the second, rightmost population was 3165.3 (95% CI: 1410.9–7101.1).

For determining the MFI-bg threshold at which a person’s sample would be considered IgG positive for anti-CHIKV antibodies, the two-component FMM has been used in previous studies with a certain number of standard deviations (SDs) added to the mean of the first component to provide this assay signal threshold [26]. For this study’s data, Figure 2 displays vertical lines that represent MFI-bg thresholds when 1, 2, or 3 SDs are added to the lognormal mean of the first component. Adding only 1 SD includes much of the population from the first component, whereas adding 3 SD appears to exclude many of the high MFI-bg signal intensities, so the 2 SD criteria was chosen as the seropositivity threshold to define a person’s blood sample as seropositive to anti-CHIKV IgG. This generated a seroprevalence estimate of 2.7% for the population (whereas 1 SD would have provided an estimate of 14.2%, and 3 SD would have given an estimate of 1.4%).

### 2.3. Correlation of Seropositivity and MFI-bg Assay Signal with Age

As shown in Table 1 and Figure 3A, although the overall seropositivity for the entire sample population was only 2.7%, increases were observed in anti-CHIKV IgG carriage with age. In fact, no individuals under 10 years of age were determined to be seropositive. The age category with the highest seroprevalence was comprised by persons older than 50 years, with 7.6% of them being anti-CHIKV IgG positive (Table 1). Regression modeling for the relationship of seropositivity with the age category provided a positive relationship (slope = 1.05; *p* = 0.002) with good correlation (R^2^ = 0.81; Figure 3A). Figure 3B shows the MFI-bg assay signal by age categories. Although a consistent increase in the MFI-bg signal is seen with increasing age, the assay signal increase by age was mostly subdued, since there was such a low overall number of seropositive persons.

### 2.4. Adjusted Odds Ratio (aOR) Estimates for Anti-CHIKV Seropositivity Based on Age, Gender, and Bednet Ownership and Usage

As described in Methods, a logistic regression was performed to generate adjusted odds ratios (aORs) in an attempt to identify any covariates that would explain risk factors for CHIKV seropositivity in this nomadic population. Overall, 74.7% of the population indicated that they owned a bednet, 61.6% indicated they had actually slept under the net the previous night, and 28.8% reported they slept under a net every night. Additional net characteristics regarding insecticide treatment (age of net, etc.) were not included in this survey questionnaire. As shown in Figure 4, the only factor significantly associated with seropositivity was age (aOR =1.05, 95% CL: 1.03–1.06; *p* < 0.0001), whereas aOR estimates for gender, net ownership, and net usage covariates were all non-significant and had wide 95% CIs that included 1.0.

## 3. Discussion

The chikungunya virus (CHIKV) has likely been circulating in Africa for hundreds (or thousands) of years, and was first identified in the mid-1900s in sub-Saharan Africa [27,28,29]. It has been proposed that the CHIKV first migrated out of Africa as early as the 18th century with the expansion of colonialism and travel in and out of the continent [30]. With the introduction of the adaptable *Ae*. *albopictus* mosquito to Europe and the New World in the late 20th and early 21st century, CHIKV outbreaks arose in naïve populations that had never been exposed to the virus before [30,31]. When measuring population exposure to CHIKV outbreaks and transmission, the clinical symptomology of presenting patients has been a useful tool for predicting infection, but assays for circulating viremia and serological indicators are also widely-used as robust metrics.

CHIKV infection induces a robust adaptive immune response, with high levels of circulating antibodies that are elicited and can be detected and quantified [32]. In this report, the CHIKV E1 antigen was used to measure past (or potentially current) exposure to the virus, and this antigen has been used in numerous serostudies in the past as a reliable marker [12,33,34,35]. In determining which blood samples were IgG positive for antibodies against the CHIKV E1 antigen, a two-component finite mixture model (FMM) was employed [26], as this method was shown to clearly define two sub-populations of IgG signal intensities in the sample population. Persons having a positive assay signal had MFI-bg values generally well above the background signal (or noise) found when assaying true negative blood samples. The FMM approach for CHIKV serology is slightly different than the serological approach for determining a positivity threshold using a panel of known negative blood samples [12,15,33]; however, a clear discrimination between the seropositive and seronegative assay signals (as was observed in the current study) means that multiple methods for defining a seropositivity threshold would provide similar seroprevalence estimates [36].

Recently, a serological study was conducted in southern Mali with school children aged 3 to 17; only 6.2% of those tested had positive IgG responses to the recombinant CHIKV E1 antigen, but increasing anti-CHIKV IgG levels were noted with increasing age [33]. However, none of these Malian schools were in the western Kayes region of Mali, which directly borders Senegal. As of early 2019, no reports could be found that have identified chikungunya in the northeast bordering nation of Mauritania, although the *Ae. aegypti* mosquito is known to be endemic there [37]. Additionally, in Senegal, CHIKV studies have tended to be concentrated in the sparsely populated southeastern region of Kedougou, bordering Mali [19], meaning that no estimates are available for the northern regions of Senegal or the nomadic pastoralist population. A previous serology-based study in southeastern Senegal in 2009 found that 2.9% of elementary schoolchildren had anti-CHIKV antibodies [19], and multiple studies have found high percentages (>75%) of monkeys that were seropositive in this region [18,19]. Although we present seroprevalence estimates in the current report, one limitation to our study is that the nomadic pastoralist population is, by definition, a mobile population, and any previous CHIKV exposure could have occurred in any of a number of areas of Senegal. The absence of seropositivity among persons 10 years of age or younger indicates that there have been no recent CHIKV outbreaks in this pastoralist population. The slow but steady increase in seroprevalence with age is more indicative of a very low-level population exposure over time, rather than dramatic outbreaks where persons of all ages are simultaneously exposed [12].

We found no evidence that bednet ownership or usage (i.e., used the previous night or every night) affected seropositivity. There are likely multiple reasons for this finding in the nomadic pastoralist population. First, there were very few seropositive persons in which to model for covariates. Secondly, although 62% of this population indicated they ever sleep under a net, *Ae. aegypti* is well known to be most active in early morning (around daybreak) and early evening [38,39], times when humans typically do not sleep. Thirdly, almost all persons that were found to be anti-CHIKV IgG seropositive were over the age of 15 years, and mass bednet distribution campaigns in Senegal have only occurred in the past decade, meaning that if there was a true protective benefit of the nets, older persons would not have had that benefit until just recently. Although insecticide-treated bednets have been shown to provide some protection against *Ae. aegypti* exposure in other human populations [40,41], the effectiveness of bednets to prevent CHIKV exposure is not well studied and should be investigated further.

As this was a mobile pastoralist population, it would be impossible to ascertain where CHIKV exposure would have occurred for seropositive persons. Additionally, this study was also limited by using a single indicator for CHIKV exposure (E1 antigen), and it is unclear how long IgG antibodies against this antigen would remain following an infection. Future seroepidemiological studies of the CHIKV may benefit from using additional antigens from the virus to assess breadth of IgG response.

In summary, a 2014 survey of nomadic pastoralists residing in northeastern Senegal found minimal evidence for exposure to CHIKV, with clear increasing trends of lifetime exposure as persons aged. Serological data provide a powerful indicator of infectious disease exposure, rather than active infection status, and can be used to generate estimates of exposure when transmission of pathogens is known to be very low. Future arbovirus studies in Senegal should aim to generate exposure estimates for populations outside of southeastern Senegal.

## 4. Materials and Methods

### 4.1. Ethics Statement

Prior to the field activities, the project was discussed with community members, health post chief nurses, and district medical officers. Participants or guardians were requested to provide written informed consent prior to their enrolment. This study was approved by the ethical review committee of the Ministry of Health, Senegal (Approval No. 324/MSAS/DPRS/CNERS, 26 August 2014).

### 4.2. Study Population, Survey, and Dried Blood Spot Sample Collection

As described previously [25], this study was carried out from September to October 2014 in the Ferlo Desert and the Senegal River Valley. Study sites known to host large numbers of nomadic pastoralists during the rainy season (primarily because of the presence of water sources for their livestock) were selected purposively. In each study area, three health districts (administrative districts) were chosen: Linguere, Ranerou, and Kanel in the Ferlo Desert; and Dagana, Podor, and Pété in the Senegal River Valley. In the Ferlo Desert, the communities of Barkédji (Linguere), Salalatou (Ranerou), and Namary (Kanel) were selected, and in the Senegal River Valley, Niassanté (Dagana), Namarel (Podor), and Ndiayéne Peul (Pété) were chosen. Nomadic pastoralists aged 6 months and older, who were returning or had just returned to the north from the south during the rainy season were recruited (the duration of the travel between north and south may last several months). Study participants reported no acute illness, had an axillary temperature <37.5 °C, and provided written informed consent before being enrolled in the study. For children, informed consent was provided by their parents or legal guardian. Potential participants were excluded if they lived permanently in the district, were not members of the nomadic community, had an acute febrile illness, or had previously taken part in the study.

A snowball sampling survey using a modified respondent-driven sampling methodology was conducted as described above. Chief nurses who worked at the health posts that served each community and were familiar with them selected leaders in the nomadic community to serve as initial seeds that would recruit other participants. Each seed was asked to recruit three other individuals, including one participant under the age of 15 years so as to include children in the sample. While this strategy made the sample more representative of the target population, it made it impossible to use classic respondent-driven sampling methodology and analysis, which depends on each participant having an independent social network from which they can recruit. We did not discourage recruitment of household members. Interviews were held at a designated central location chosen to facilitate participation. Each subsequently recruited participant received the same instructions, with recruitment continuing until the sample size of 1800 was reached (300 in each of the six sites). To calculate the sample size, a target proportion of 50% for key variables such as care seeking and net use with a confidence level of 95% and a confidence interval of (0.40, 0.60), and a design effect of 3 were used, giving 300 participants per site and a total of 1800 participants.

From each individual, whole blood by finger prick was spotted onto Whatman 903 Protein Saver Cards or Whatman 1 circular filter paper (GE Healthcare, Piscataqway, NJ, USA) and allowed to air dry for at least 2 h. The dried blood spots (DBS) were individually packaged into plastic bags with desiccant, and stored at 4 °C prior to analysis. Of 1800 participants, 1465 (81.4%) had a DBS available for serological lab assays.

### 4.3. CHIKV E1 Antigen Coupling to Beads

Carboxyl groups on the surface of spectrally classified-magnetic polystyrene microspheres (MagPlex Beads; Luminex Corporation, Austin, TX, USA) were converted to reactive esters using the 1-ethyl-3-(3-dimethylaminopropyl) carbodiimide method (Calbiochem, Woburn, MA, USA). Then, 8.7 µg of recombinant CHIKV E1 antigen (CTK Biotech, San Diego, CA, USA) was covalently linked to microspheres (12.5 million) by covalent amide bonds in phosphate buffered saline (PBS), pH 7.2, through the reaction of amino groups on the antigen and the created ester groups on the beads. Serum specimens known to be highly reactive to the CHIKV were used to determine coupling efficiency as described previously [12,33].

### 4.4. Sample Processing, Blood Elution, and Anti-CHIKV E1 IgG Immunoassay

A 6-mm circular punch corresponding to 10 μL of whole blood from the center of each DBS was placed in 200 μL buffer (PBS containing 0.5% polyvinyl alcohol (Sigma, St. Louis, MO, USA), 0.8% polyvinylpyrrolidine (Sigma, St. Louis, MO, USA), 0.1% casein (ThermoFisher Scientific, Waltham, MA, USA), 0.5% bovine serum albumin (Millipore, Burlington, MA, USA), 0.3% Tween-20, 0.05% sodium azide, and 0.1% crude and unclarified *Escherichia coli* extract (for absorption of *E. coli* antibodies for coupled antigens expressed in *E. coli*) and incubated at 4 °C at least overnight.

For the IgG detection assay, wash steps occurred with 0.05% Tween 20 in PBS using a hand-held magnet appropriate for 96-well plates (Luminex Corp, Austin, TX, USA). In 5 mL Buffer A, a bead mix was prepared including the beads coupled to CHIKV E1, and 50 µL of bead mix was pipetted into each well of the assay plate (BioPlex Pro plates, Bio-Rad, Hercules, CA, USA). Approximately 625 CHIKV E1 beads were added per well. Beads were washed 2×, and 50 µL of reagent mix (in 5 mL Buffer A: 1:500 anti-human IgG (Southern Biotech, Birmingham, AL, USA), 1:625 anti-human IgG_4_ (Southern Biotech, Birmingham, AL, USA), and 1:200 streptavidin-PE (Invitrogen, Waltham, MA, USA)) was added to all wells. Then, 50 µL samples at 1:50 dilution (or controls) were added to the reagent mix in the appropriate wells. Plates were incubated overnight with gentle shaking at room temperature and protected from light. The next morning (after ~16 h of total incubation time), plates were washed 3×, and beads were resuspended with 100 µL PBS and read on the MAGPIX® machine. Data were acquired with a MAGPIX® reader using xPONENT® software (Luminex Corp, Austin, TX, USA), which calculated the integrated intensity of the sum of all pixels generated from wavelength-specific light emitting diodes (LED), determining bead classification and median fluorescence intensity (MFI) from the reporter molecule, using the image of a target of 50 magnetically captured beads. Background (bg) fluorescence from a blank with no DBS was subtracted (MFI-bg), and this difference was used as data for analyses. If samples had a high value or did not provide a value for the generic glutathione-*S*-transferase (GST) antigen, which served as an internal non-binding control for this assay, they were removed from analysis. This was the case for two samples in the study.

### 4.5. Statistics

Statistical analyses were performed in SAS 9.4 (SAS Institute, Cary, NC, USA). Log-transformed MFI-bg values were fit to a two-component finite mixture model by the FMM procedure with normal distribution and maximum likelihood estimation outputs for component means and variances. A seropositivity threshold was determined by the mean + standard deviations of the first (putative seronegative) component.

Dynamics of seropositivity to CHIKV IgG by age category were modeled by the GLM procedure with the MODEL statement with alpha of 0.05. The generation of adjusted odds ratio (aOR) estimates for relationship between IgG against CHIKV E1 antigen and age, net ownership of the respondent, net use the previous night, and net usage every night was modeled with logistic regression (LOGISTIC procedure, MODEL statement) and confidence intervals reported as 95% Wald CIs with a null hypothesis of *β*_x_ = 0. The logistic regression model was written as
Logit P (chikpos =1) = *β*_0_ + *β*_1_(age(years)) + *β*_2_(net ownership) + *β*_3_(net use previous night) + *β*_4_(net use every night) + *β*_5_(sex).

## Figures and Tables

**Figure 1 pathogens-08-00113-f001:**
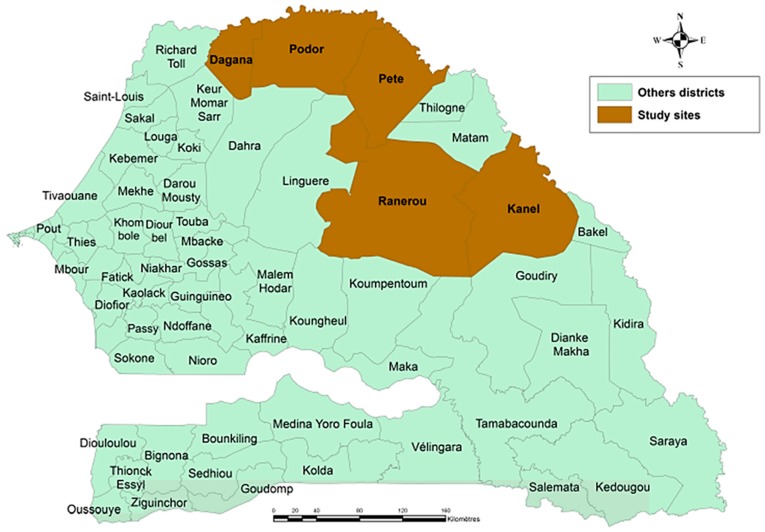
Senegalese districts included in the nomadic pastoralist study. Persons of all ages were sampled from five districts in Senegal from September to October 2014.

**Figure 2 pathogens-08-00113-f002:**
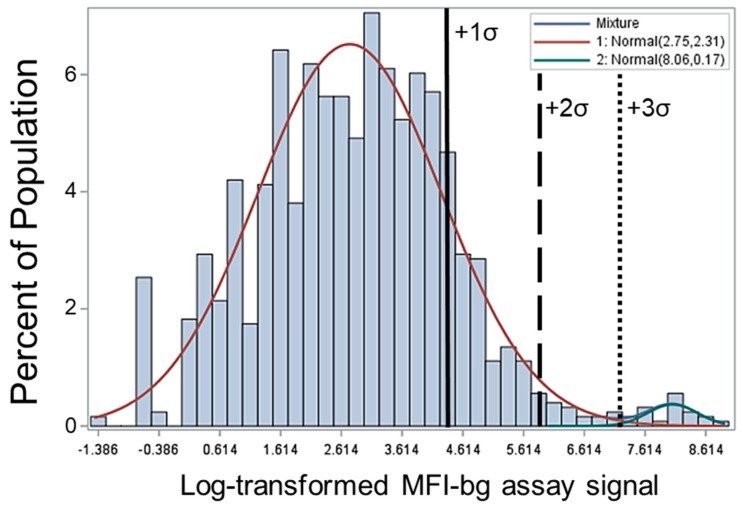
Distribution of assay signal intensity for IgG antibodies against chikungunya of nomadic pastoralist populations in Senegal, 2014. Log-transformed median fluorescence intensity minus background (MFI-bg) assay signal values are shown, modeled with a two-component finite mixture model. For each component, log-transformed mean and variance are shown in the inset. Vertical lines show seropositivity cutoff thresholds when defining the assay signal threshold as lognormal mean plus 1, 2, or 3 standard deviations (solid, long hash, and short hashed lines, respectively).

**Figure 3 pathogens-08-00113-f003:**
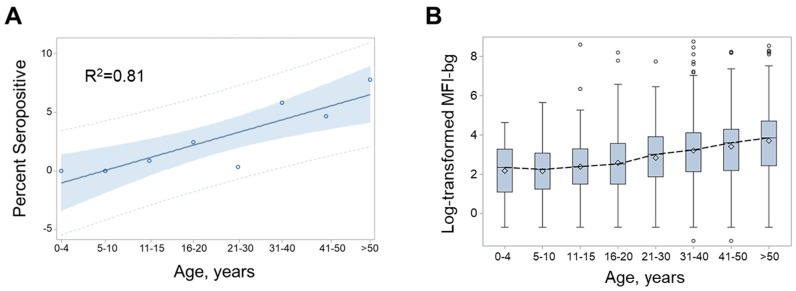
Anti-chikungunya IgG seroprevalence or assay signal by age for nomadic pastoral populations in Senegal, 2014. (**A**) Linear regression model for age categories plotted by percent seropositivity. Plot shows circles for percent of each age category seropositive for CHIKV, and a linear regression line of the best fit with 95% confidence limits in shading, and 95% confidence intervals as hashed lines; R-square value for regression model is displayed in the inset. (**B**) Log-transformed assay signal data is shown as boxplots for the log-transformed median fluorescence intensity minus background (MFI-bg) assay signal by age category. Boxplots are displayed as interquartile range (IQR) with whiskers extending 1.5× above and below IQR and circles for observations beyond this. A dashed line connects boxplot medians.

**Figure 4 pathogens-08-00113-f004:**
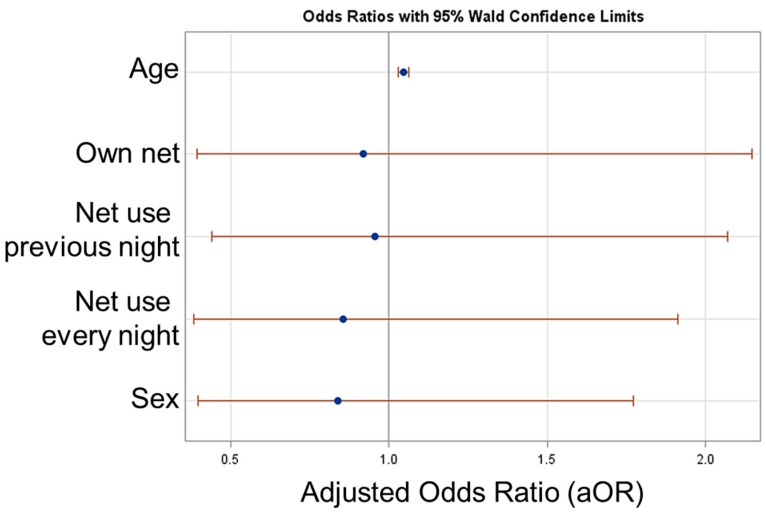
Adjusted odds ratio (aOR) estimates for likelihood of chikungunya seropositivity for age, bednet ownership, bednet usage, and gender of nomadic pastoral populations in Senegal, 2014. Plot displays aOR point estimates with 95% confidence intervals. Age was kept continuous, and for the variable “Sex”, female was coded as “1” and male as “0”.

**Table 1 pathogens-08-00113-t001:** Number of participants enrolled and chikungunya seroprevalence by age category of nomadic pastoral populations in Senegal, 2014.

Age Category	Number Enrolled	Number Seropositive (%; 95% Confidence Interval)
0–4	112	0 (0.0%; −2.9–1.3%)
5–10	188	0 (0.0%; −1.5–1.9%)
11–15	172	2 (1.2%; −0.1–2.7%)
16–20	190	5 (2.6%; 1.1–3.5%)
21–30	295	3 (1.0%; 2.2–4.6%)
31–40	186	11 (5.9%; 3.1–5.8%)
41–50	133	6 (4.5%; 3.8–7.2%)
>50	158	12 (7.6%; 4.4–8.6%)

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
