# Peer review of "Serological Data Shows Low Levels of Chikungunya Exposure in Senegalese Nomadic Pastoralists"

_pathogens, 2019, doi:10.3390/pathogens8030113_

Round 1

Reviewer 1 Report

Line 44, please cite original paper (PMCID: PMC2218030, PMID: 13346078) instead of a review citation #7.

Line 102, correct as anti-‘CHIKV’

Line 139, Figure 4. Legend should read as ‘bednet’ ownership

Did the questionnaire capture information on individuals travel history to the South where CHIKV is endemic? If yes then that should also be factored in logistic regression model.

Author Response

Line 44, please cite original paper (PMCID: PMC2218030, PMID: 13346078) instead of a review citation #7.

Thank you for the suggestion, and this has been replaced.

Line 102, correct as anti-‘CHIKV’

Corrected

Line 139, Figure 4. Legend should read as ‘bednet’ ownership

Corrected

Did the questionnaire capture information on individuals travel history to the South where CHIKV is endemic? If yes then that should also be factored in logistic regression model.

The reviewer brings up a keen point here, but unfortunately, information about travel history to the south was not collected in this questionnaire.

Reviewer 2 Report

This study was conducted in Senegal to determine seroprevalence for chikungunya among nomadic Senegal population. The results could be of great value to add to our understanding of historic and recent spread of chikungunya. While the study seems to have been well designed and written, there is one important point that needs clarity.

The survey design was based on a log-normal population and sero-positivity was based on mean +2SD of log- assay signal. However, this design assumes the mean signal represents negative controls. How did you decide that this was a population of negatives and thus those with the most extreme (+2SD) signals were positives? Could it not be the case that the population is for positives and thus the extreme small signals are for negatives? I could not figure out which is valid based on the evidence the authors seem to have relied on. 

Other minor comments:

1.    What are health districts? Are these administrative units or some forms of units created for the purpose of delivery of health care services?

2.    If they are nomadic how would you attribute their seroprevalence to their (or any) location?

3.    Line 293: The final logistic regression model was written as: … How was the final GLM model reached at?

Line 86-87: The lognormal MFI-bg mean for the second, rightmost population was 313165.3 (95% CL: 1410.9-7101.1).  This value does not make sense.

Author Response

This study was conducted in Senegal to determine seroprevalence for chikungunya among nomadic Senegal population. The results could be of great value to add to our understanding of historic and recent spread of chikungunya. While the study seems to have been well designed and written, there is one important point that needs clarity.

The survey design was based on a log-normal population and sero-positivity was based on mean +2SD of log- assay signal. However, this design assumes the mean signal represents negative controls. How did you decide that this was a population of negatives and thus those with the most extreme (+2SD) signals were positives? Could it not be the case that the population is for positives and thus the extreme small signals are for negatives? I could not figure out which is valid based on the evidence the authors seem to have relied on. 

The reviewer brings up an astute point here regarding dichotomization of the signal intensity data from the IgG detection assay, and the answer is multifactorial. In the absence of an international serological standard for objective determination, this same question would be posed for any quantitative immunoassay: “what assay signal constitutes an IgG true positive for X antigen”. After log-transformation of the assay signal data, the 2-component finite mixture model (FMM) has been used by many disease groups in order to attempt to find the true seronegative (lower signal) and seropositive (higher signal) sub-populations. In this study, we try to emphasize how well statistically (and visually) that the 2-component FMM fit to the Senegalese data, and that a clear bimodal distribution of assay signal could be observed. The mean+2SD MFI-bg assay signal threshold was determined to be 327 for this current study, whereas it was 594 and 640 from previous studies in our group (cited in the paper) which have used “non-exposed US residents” as the true seronegative population. On a MFI scale from 0 to 32,000 for the Luminex platform, the three thresholds listed above are quite similar.

With all of this said, it is still good to have some reference to what CHIKV IgG seroprevalence should be in this area of the world. With previous human seroestimates in (the more tropical) southeast Senegal and western Mali being under 10%, the strong hypothesis would be that persons spending most of their time in northern Senegal would be largely unexposed to this arbovirus, and likely under 10% as well. So if only using a signal threshold of the mean+1SD of this putative seronegatives (left component), the seropreveance in the nomadics jumps to over 14%, which seems high, and not supported by the FMM fitting of the data. We present seroprevance estimates in the text for mean+1SD, +2SD, and +3SD for the reader’s information, but we felt that the +2SD threshold criteria was supported by both the FMM fitting to the MFI data as well as recent anti-CHIKV IgG estimates in surrounding areas.       

Other minor comments:

1.    What are health districts? Are these administrative units or some forms of units created for the purpose of delivery of health care services?

 - Yes, these are administrative districts, and this additional text for clarifying has been added to Line 218 in Methods.

2.    If they are nomadic how would you attribute their seroprevalence to their (or any) location?

 - The reviewer raised a great point here in that lasting IgG carriage has no stamp on where exposure would have taken place, and we raise this as a study limitation in Discussion. For this report, we include a map to illustrate where the pastoralists were sampled from at time of study enrollment, but we make no assumptions or predictions to where they would have been exposed to CHIKV. In this study, we attempt to estimate the exposure to CHIKV for this nomadic population as a whole, and do not attempt to decipher where that exposure would have taken place.   

3.    Line 293: The final logistic regression model was written as: … How was the final GLM model reached at?

 - As written, the authors are not exactly sure the nature of the first half of the question above. The GLM approach was used for the modeling exercise of CHIKV seropositivity only with increasing age because of the linear-response nature of this relationship (Fig 3A) and a constant change in the predictor leading to a constant change in the response variable. For modeling with multiple covariates, logistic regression was chosen to account for multiple variables and the nature of relationships not known.    

Line 86-87: The lognormal MFI-bg mean for the second, rightmost population was 313165.3 (95% CL: 1410.9-7101.1).  This value does not make sense.

The reviewer correctly points out a typo in the text, and the number has been corrected to 3165.3

Reviewer 3 Report

The article by Seck et. al. describes the results of a serological study to evaluate Chikungunya virus (CHIKV) exposure in Senegalese Nomadic Pastoralists. Although CHIKV is known to be present in Senegal, this is the first study to examine population exposure to CHIKV in Senegalese nomadic populations. There are some limits to the study, but the authors adequately address these in the discussion.

Specific comments:

(1) Lines 85-87 state that the lognormal MFI-bg means are 15.6 and 313165.3 for the leftmost and rightmost populations, respectively. However, line 92 in the Figure 2. legend states that the lognormal means are shown in the inset and these numbers are not the same as those in the text. Please review and revise as necessary.

(2) Line 110: After "positive" add "(Table 1)".

(3) Recommend presenting Table 1. before Figure 3., since the text describes data for Table 1 prior to data for Figure 3.

(4) Figure 3 legend: Please define the circles present in Fig. 3A.

(5) Line 179: delete "were", so that it reads "...age or younger indicates..".

(6) Line 197: delete "as", so that it reads "...indicator for CHIKV exposure being...".

(7) Line 217: add "the" following "during", so that it reads "pastoralists during the rainy season...".

Author Response

The article by Seck et. al. describes the results of a serological study to evaluate Chikungunya virus (CHIKV) exposure in Senegalese Nomadic Pastoralists. Although CHIKV is known to be present in Senegal, this is the first study to examine population exposure to CHIKV in Senegalese nomadic populations. There are some limits to the study, but the authors adequately address these in the discussion.

Specific comments:

(1) Lines 85-87 state that the lognormal MFI-bg means are 15.6 and 313165.3 for the leftmost and rightmost populations, respectively. However, line 92 in the Figure 2. legend states that the lognormal means are shown in the inset and these numbers are not the same as those in the text. Please review and revise as necessary.

- The reviewer raises a good point here, and the authors should have been more specific with their language usage. The text in the main body will remain the same, but for the figure legend, the text has been changed from “lognormal mean and variance” to “log-transformed mean and variance”

*The 313165.3 number is actually a typo, and has been corrected to 3165.3

(2) Line 110: After "positive" add "(Table 1)".

- Added

(3) Recommend presenting Table 1. before Figure 3., since the text describes data for Table 1 prior to data for Figure 3.

- The authors would agree with this suggestion, but it would ultimately be at the discretion of the editors.

(4) Figure 3 legend: Please define the circles present in Fig. 3A.

- Additional text has been added to the legend to further define.

(5) Line 179: delete "were", so that it reads "...age or younger indicates..".

- Deleted

(6) Line 197: delete "as", so that it reads "...indicator for CHIKV exposure being...".

- Deleted

(7) Line 217: add "the" following "during", so that it reads "pastoralists during the rainy season...".

- Added

Round 2

Reviewer 2 Report

The authors have answered most questions to my satisfaction.  I have one clarifying point:

Line 293: The final logistic regression model was written as: … How was the final GLM model reached at?

 - As written, the authors are not exactly sure the nature of the first half of the question above. The GLM approach was used for the modeling exercise of CHIKV seropositivity only with increasing age because of the linear-response nature of this relationship (Fig 3A) and a constant change in the predictor leading to a constant change in the response variable. For modeling with multiple covariates, logistic regression was chosen to account for multiple variables and the nature of relationships not known.    

Response:

What I am suggesting is that since you mentioned the model to be a final model (it suggests you started of with something else), how was the final model structure selected? In other words, how did you decide which covariates to include? Did you do some way of comparing different forms based on goodness of fit? If, yes explain. If no, how would you justify using this form? 

Author Response

Thank you for the further explanation of this previous comment.

For the GLM approach (Fig 3A, slope and R^2 estimates), only the direct relationship between seropositivity by age was investigated, so only age is included in this GLM model.

For the logistic regression (Fig 4 and estimates), the authors realize the inclusion of the word "final" here is misleading since we did not go through a model selection process. Due to the very limited questionnaire administered during the survey, the covariates included in the model were the only ones available, and thus, all available covariates were included in the logistic regression model. The word "final" will be removed here.

* The authors realize the covariate "Sex" is not currently written into the model in Line 293, and this will be added as well